# Study on the Preparation Method of Quality-Assured In-Hospital Drug Formulation for Children—A Multi-Institutional Collaborative Study

**DOI:** 10.3390/children10071190

**Published:** 2023-07-10

**Authors:** Jumpei Saito, Eiji Suzuki, Yosuke Nakamura, Takashi Otsuji, Hiroshi Yamamoto, Hideki Yamamoto, Yuiko Kai, Maiko Totsu, Sayuki Hashimoto, Hidefumi Nakamura, Miki Akabane, Akimasa Yamatani

**Affiliations:** 1Department of Pharmacy, National Center for Child Health and Development, Tokyo 157-8535, Japan; 2Department of Pharmacy, Nagano Children’s Hospital, Nagano 399-8288, Japan; 3Department of Pharmacy, Shiga Medical Center for Children, Moriyama 524-0022, Japan; 4Department of Pharmacy, NHO Shikoku Medical Center for Children and Adults, Zentsuji 765-8507, Japanyamamoto.hideki.qc@mail.hosp.go.jp (H.Y.); 5Department of Pharmacy, Kanagawa Children’s Medical Center, Yokohama 232-0066, Japan; 6Department of Pharmacy, Aichi Developmental Disability Center, Kasugai 480-0392, Japan; 7Department of Research and Development Supervision, National Center for Child Health and Development, Tokyo 157-8535, Japan; 8Pediatric and Perinatal Pharmacology, Meiji Pharmaceutical University, Tokyo 204-0004, Japan

**Keywords:** pediatric formulation, oral powder, stability, quality assurance

## Abstract

The quality-assured preparation of crushed and diluted preparations for children is a challenge. In this study, a multicenter study was conducted to validate the preparation method for the quality assurance of baclofen powder, clonidine powder, and hydrocortisone powder prepared from tablets according to a previously established method. In-hospital preparations were prepared at five medical facilities under different crushing and mixing conditions. After storage in closed bottles, in-use bottles, and laminated paper for 120 days, ingredients stability, drug elution, and content uniformity after packaging were evaluated. All three ingredients were maintained at between 90% and 110% of their initial content for 120 days under packaging conditions of 25 ± 2 °C and 60 ± 5% relative humidity, with no change in dissolution in all formulations made at all five facilities. The content uniformity was also acceptable. The established method may contribute to quality-assured pediatric dosage form modification.

## 1. Introduction

In pediatric medical practice in Japan, tablets are often crushed, or capsules decapsulated in the pharmaceutical department of medical facilities when oral formulations are not flexible for dose adjustment or when tablets cannot be administered as supplied without modification [1,2,3]. With the increasing demand for drug information, information on the stability of tablets after crushing is sometimes provided by the pharmaceutical companies that manufacture and sell the tablets. On the other hand, in actual medical practice, tablets are crushed or decapsulated prior to dispensing as an in-hospital preparation, rather than being crushed when necessary. In pediatric patients, where the number of crushed and decapsulated tablets is high in relation to the total number of oral prescriptions, in-hospital preparation as a ready-to-use product is essential to improve operational efficiency [1]. Many in-hospital pre-products often lack quality assurance, such as the assurance of long-term stability and ingredient dissolution. The National Center for Child Health and Development (NCCHD) has been studying the standardization of preparation methods to ensure the quality of in-hospital preparation for pediatric drug therapy, and has examined preparation methods for baclofen, hydrocortisone, and clonidine [4,5,6]. Although the quality of the preparations prepared at the Department of Pharmacy of NCCHD was confirmed to be assured, the quality of products prepared in different environments and by different methods at other facilities is unclear. The purpose of this study was to confirm the robustness and validity of the standard protocol for in-hospital formulations by evaluating the quality of premixed products prepared at medical facilities other than NCCHD using the published preparation methods or under different preparation conditions. Additionally, in particular, the uniformity of drug content after packaging is required because powder medicines in Japan are generally packaged in single doses by automatic packaging machines and provided to patients in sachets. Although there have been no detailed physical properties studies on the post-packaging uniformity of powders, it is generally reported that in the pharmaceutical manufacturing process, the particle size of a powder affects the uniformity of the composition of the main ingredients and excipients when adhering to equipment and filling the equipment [7,8,9,10]. Therefore, different crushing and mixing conditions may affect uniformity after dispensing and during weighing. We attempted to conduct an additional study on this Japan-specific issue of uniformity after packaging as a particularly important research item because of its impact on clinical efficacy.

## 2. Methods

### 2.1. Preparation Method

To prepare the powder formulations, Gabalon^®^ 10 mg 100 tablets (Alfresa Corporation, Osaka, Japan) was used for baclofen powder, and Catapres^®^ 0.075 mg 100 tablets (Medical Parkland K.K., Tokyo, Japan) was used for clonidine hydrochloride. Hydrocortisone was supplied by Cortril Tablets 10 mg 100 tablets (Pfizer Japan Inc., Tokyo, Japan). After crushing and sieving, the product was diluted to 10 mg/g, 0.2 mg/g, and 20 mg/g, respectively, using crystalline lactose (EFC Lactose Whey, Viatris Inc., Tokyo, Japan) for dilution. The crushing and mixing conditions of each preparation at each of the five institutions are shown in Table 1. An automatic tablet mill was used for tablet crushing at all five medical institutions. Two facilities each used the same crushing conditions, either 14,500 rpm or 6000 rpm. The remaining facility crushed at 16,320 rpm. The crushed tablets were sieved through a sieve with a 300 µm (one facility) or 500 µm (four facilities) aperture. Manual mixing with a pestle and mortar was performed at two medical facilities and mixing with an automatic mixer was used at three facilities.

### 2.2. Storage Conditions

The prepared in-hospital pre-products were stored under the following three storage conditions according to the previous study [4,5,6,11]: (1) “Bottle (closed)” condition: stored in polycarbonate bottles (Yamayu, Osaka, Japan), (2) “Bottle (in-use)” condition: 0.1 g weighed daily from the bottle condition, and (3) package condition: packaged in cellophane and polyethylene wrapping paper (TK-70W, Takazono, Tokyo, Japan), and then stored under light-shielded conditions at 25 °C ± 2 °C and 60% ± 5% relative humidity (RH). The samples were stored for 0 to 120 days.

### 2.3. Stability Tests

For drug stability, the samples were obtained from each storage condition at days 0, 30, 60, 90, and 120, diluted according to previously published methods [4,5,6] and analyzed using an HPLC system Ultimate 3000 (Thermo Fisher Scientific, Tokyo, Japan) containing an autosampler, a column oven, and a diode array detector (Table 2). The concentration changes were calculated as (measured concentration/initial concentration) × 100 (%), and within 10% of the initial concentration were considered acceptable changes [12].

### 2.4. Assay Validation

Linearity was determined by triplicate and the concentration ranges were set as follows: 2.0 to 20.0 μg/mL for baclofen, 0.01 to 1.0 μg/mL for baclofen impurities, 2.0 to 20.0 μg/mL for clonidine hydrochloride, 0.01 to 1.0 μg/mL for 2,6-DCA, 0.5 to 100 μg/mL for hydrocortisone, and 0.05 to 10 μg/mL for 10 hydrocortisone impurities; six concentration levels were set for all drugs. Validation was performed to ensure that sample concentrations were within the linear analyte response range. Calibration standard concentrations were back-calculated to verify the suitability of the standard calibration curves. Standard calibration curves and correlation coefficients were obtained using a weighted linear regression method (weighted coefficient: 1/x^2^). The calibration curve was considered acceptable for measurement with the LC-DAD system if the correlation coefficient was 0.99 or higher. The accuracy was then evaluated in terms of reproducibility (within day) and intermediate accuracy (between day). Data were presented as mean values and relative standard errors. Reproducibility was evaluated by six repeated injections of a standard solution spiked at 10 μg/mL. Intermediate precision was assessed with six repeat injections on three different days. Accuracy was calculated as (measured concentration/nominal concentration) × 100 (%). Statistical analysis was performed using one-way analysis of variance and limits of detection and quantification calculated using ICH guidelines. Each calibration curve was stored at −80 °C, as these samples were also used for quality control (QC). The standard deviations of the response (Sy) of the calibration curve and the slope (S) of the calibration curve were used to determine the lower limit of detection (LOD). The standard deviation of the response was determined from the standard deviation of the y-intercept of the regression line. The lower limit of quantitation (LLOQ) was obtained by multiplying Sy/S by 10.

### 2.5. Detection of Impurities

In accordance with previous studies, baclofen impurities A and B [13] for baclofen, 2,6-dichloroaniline (2,6-DCA, also called clonidine degradation product) [14] for clonidine hydrochloride, and hydrocortisone impurities (A, B, C, D, E, F, G, H, I, N) for hydrocortisone were examined [15].

### 2.6. Drug Dissolution Tests

Dissolution tests were performed under the same conditions as previously reported [4,5,6] according to ICH Japanese Pharmacopoeia 17.6.10. (paddle method; NTR-6400AC; Toyama Sangyo, Tokyo, Japan) using 900 mL of dissolution solution stirred at 37 °C ± 0.5 °C and 50 rpm. Mean dissolution rates were compared to those of day 0 formulation.

### 2.7. Uniformity of Drug Product Content Test

Since the crushing conditions at each medical institution were different, the particle size of the prepared formulation would be different, which may affect the uniformity of drug content after the packaging of the powder. To confirm the uniformity of the content of the drug product, we used an automatic dispersing machine (Crestage-Pro2, Takazono, Tokyo, Japan) to divide each package into 30 packages, each containing 0.5 g. Ten of these packages were then taken and the content of the ingredients was determined. The determination value was then calculated according to the following formula, and if the acceptance value did not exceed 15.0 the product was considered to be acceptable [16].
Acceptance value = |M − X| + ks(1)

X, average of the individual content expressed as a percentage of the indicated amount (5 mg for baclofen, 0.1 mg fir clonidine, and 10 mg for hydrocortisone). k, Determination coefficient (2.4). s, Standard deviation. M, M = X when 98.5% ≤ X ≤ 101.5%, M = 98.5% when X < 98.5%, M = 101.5% when X > 101.5.

## 3. Results

The results of the ingredient stability tests, changes in the dissolution properties of each formulation, and the uniformity of drug contents after packaging made at five medical facilities with different crushing and mixing conditions were shown.

### 3.1. Stability Study

The ingredient stability of the in-hospital preparations of the study drugs, baclofen, clonidine hydrochloride, and hydrocortisone, was examined under each of the three storage conditions (closed bottle, in-use bottle, and packaging) after being prepared at the five medical facilities and stored for 120 days. The ingredients content was within specification, ranging from 90.0% to 110.0% of the initial concentration (Table 3) [12].

### 3.2. Impurities Studies

Baclofen impurities A and B [13] and 2,6-DCA (Clonidine impurity-C), an impurity for clonidine hydrochloride [14], were not detected in the samples after any of the storage conditions or the storage period. For hydrocortisone, small amounts of impurities B (cortisone) and G (hydrocortisone-21-aldehyde) were found in all preparations prepared at the five medical facilities, regardless of storage conditions, in samples taken on day 120 as previously reported [6], but relative concentrations were less than 0.05% of hydrocortisone. No other impurities were detected.

### 3.3. Dissolution Tests

Dissolution profiles of all components were comparable to those of day 0 immediately after preparation under all storage conditions for the formulations prepared at all five medical facilities, and no changes were observed with long-term storage (see Appendix A).

### 3.4. Uniformity of Drug Content Test

The formulations prepared at each medical institution were divided into 30 packets, and the component content of 10 of these packets was quantified and a determination value was calculated. The content uniformity values for each ingredient after preparation at the five institutions were all less than 15 (Table 4).

## 4. Discussion

The results of this study suggest that the standard protocols discussed in this study will enable the provision of quality-assured in-hospital preparations of baclofen, clonidine hydrochloride, and hydrocortisone to pediatric patients.

The in-hospital preparations were prepared at five medical facilities under different crushing and mixing conditions, and all three ingredients remained stable up to 120 days, similar to the preparations made in the Department of Pharmacy at the NCCHD. The generation of impurities was also similar. No change in dissolution was observed after 120 days of storage. In our previous study of the quality evaluation of in-hospital formulations, the uniformity of formulation content after dispensing in a dispensing machine was not sufficiently evaluated. In this study, we conducted a formulation content uniformity test in accordance with the regulations stipulated in the Japanese Pharmacopoeia, and were able to confirm that the formulation met the criteria for content uniformity after packaging. Uniformity was maintained after packaging, regardless of the crushing method.

In Japan, tablet crushers for crushing tablets and mixers or mortars for mixing crushed powder are commonly available for changing the dosage form for pediatric use, and it is considered possible to prepare in-hospital formulations with assured quality by using the method described in this study. Although the storage conditions were extremely limited, to 25 °C ± 5 °C and 60% ± 5%RH in a light-shielded environment, this is considered feasible in a clinical setting where pediatric care is available.

All three formulations studied in this study are frequently crushed and administered to the pediatric population. Since there are no appropriate pediatric formulations in the world, many studies have been conducted on compounding these three drugs [14,16,17,18,19,20,21,22,23,24,25], and there have been reports of errors in preparation and incidents involving patients during the compounding process [26,27,28,29,30]. Therefore, in addition to assuring quality, there is an urgent need to ensure safety through the standardization of formulation preparation. Some other countries have established protocols for standardized preparation methods and have normalized in-hospital formulations [31,32,33,34]. The development of dose-adjustable formulations from pharmaceutical companies is highly warranted. In addition to the three drugs examined in this study, there are many other drugs that have been compounded in the pediatric setting, and it is believed that drug therapy is currently being administered without an assurance of quality (or based only on limited information provided by pharmaceutical companies). It is difficult to request pharmaceutical companies to develop all drugs. Similar studies for other drugs may be needed to continue improving the environment for drug use in pediatric pharmacotherapy.

In addition, this study did not confirm changes in the crystal system using powder X-ray diffraction and differential scanning calorimetry. At least, no significant changes in dissolution were observed in the preparation method and after long-term storage, suggesting that there were no physical changes that could affect the dissolution. Limitations of this study include the following: (1) The stability, homogeneity, and dissolution of the drug product were only verified, and it is not clear whether bioequivalence is ensured. (2) The quality may not be guaranteed when prepared with different crushing and mixing equipment and with different concentrations and raw materials (e.g., generic drugs). (3) The safety and efficacy of the drug product should be investigated separately. Efforts should continue to be made to ensure the quality of compounding in pediatric drug therapy, which is currently being conducted at each medical facility. Baclofen, clonidine, and hydrocortisone, which were selected as model drugs in this study, are crushed in many pediatric facilities in Japan, and excipients such as lactose are added and provided to patients. Similarly, they are frequently crushed, compounded in the pharmacy, divided by a dispensing machine as a long-term drug, and administered. Although only three drugs were examined in this study, there are many drugs whose quality should be ensured. Although quality standardization efforts are underway outside of Japan, we believe that the quality evaluation of more drugs will continue to be essential.

## 5. Conclusions

The results of this study suggest that compliance with the procedures established in the previous study will enable the supply of uniform and quality-assured in-hospital formulations.

## Figures and Tables

**Table 1 children-10-01190-t001:** Five facilities where in-hospital formulations were practiced and the conditions for crushing and mixing.

Facility	Crusher: Conditions	Sieve Aperture	Mixing: Conditions
I	Automatic tablet crusher (KC-HUK2, Konishi Medical Instruments Co., Ltd., Osaka, Japan): 6000 rpm, 0.5 min	500 µm *	Manual (mixing with mortar pestle), 3 min
II	Automatic tablet crusher (Smasher Model HTF-35, Daido Kako Co., Ltd., Osaka, Japan): 14,500 rpm, 0.5 min	300 µm *	Automatic mixer (MW-2, TOSHO Corporation, Tokyo, Japan): 700 revolutions/min for rotation, 20 revolutions/min for revolution, 60 s), 3 min
III	Automatic tablet crusher (KC-HUK2, Konishi Medical Instruments Co., Ltd., Osaka, Japan): 6000 rpm, 0.5 min	500 µm *	Automatic mixer (YM-500, Yuyama Manufacturing Co. Ltd., Tokyo, Japan): 750 revolutions/min for rotation, 24 revolutions/min for revolution, 60 s), 3 min
IV	Automatic tablet crusher (YM-200II, Yuyama Manufacturing Co., Ltd., Tokyo, Japan): 16,320 rpm, 0.5 min	500 µm *	Automatic mixer (YM-500, Yuyama Manufacturing Co., Ltd., Tokyo, Japan): 750 revolutions/min for rotation, 24 revolutions/min for revolution, 60 s), 3 min
X	Automatic tablet crusher (Smasher Model HTF-35, Daido Kako Co., Ltd., Osaka, Japan): 14,500 rpm, 0.5 min	500 µm *	Manual (mixing with mortar pestle), 3 min

* crushing was repeated until all were sieved.

**Table 2 children-10-01190-t002:** Assay for each target drug.

	Baclofen	Clonidine	Hydrocortisone
Autosampler temperature	10 °C	10 °C	10 °C
Column oven	40 °C	40 °C	40 °C
Separation column	C18 column (Imtakt US-C18 column; length 150 mm, inner diameter 3.0 mm, particle size 5 µm; Imtakt Co., Ltd., Kyoto, Japan)	C18 column (Imtakt US-C18 column; length 150 mm, inner diameter 3.0 mm, particle size 5 µm; Imtakt Co., Ltd., Kyoto, Japan)	C18 column (Imtakt US-C18 column; length 150 mm, inner diameter 3.0 mm, particle size 5 µm; Imtakt Co., Ltd., Kyoto, Japan)
Eluents	A: 10 mM ammonium formate solution B: 0.1% acetonitrile formate solution	A: 10 mM ammonium formate solution, adjusted to pH 4.0 with formic acid B: 0.1% formic acid in acetonitrile	A: 10 mM ammonium formate in water, adjusted to pH 3.5 B: acetonitrile
Separations	Isocratic mode with 60% B composition	Isocratic mode with 60% B composition	A gradient with two isocratic separation modes 0 to 20 min: the composition of eluent B was maintained at 40% 20 to 27 min: 60% of eluent B 27 to 32 min: the composition of eluent B was reduced to 40%
Flow rate	0.4 mL/min	0.4 mL/min	0.4 mL/min
Separation time	8 min	10 min	32 min
Injection volume	5 µL	5 µL	5 µL
Detection	220 nm	210 nm	254 nm

**Table 3 children-10-01190-t003:** Results of stability study.

Facility	Storage Conditions	Storage Conditions	Storage Container	Test Periods (Days)				
				0	30	60	90	120
Baclofen								
I	Classical	25 ± 2 °C/60 ± 5% RH	Amber/PC bottle	100.0%	100.4 ± 3.4%	99.8 ± 4.8%	99.0 ± 1.8%	99.6 ± 4.7%
	In-use		Amber/PC bottle	100.0%	99.9 ± 4.9%	98.9 ± 5.1%	99.1 ± 4.1%	98.7 ± 3.1%
	Post-package		Amber/CP laminated paper	100.0%	101.0 ± 4.4%	100.7 ± 1.3%	100.3 ± 4.7%	99.6 ± 1.3%
II	Classical	25 ± 2 °C/60 ± 5% RH	Amber/PC bottle	100.0%	100.4 ± 3.4%	100.8 ± 4.8%	99.0 ± 1.5%	99.6 ± 4.6%
	In-use		Amber/PC bottle	100.0%	99.0 ± 4.9%	98.9 ± 1.1%	99.1 ± 4.4%	98.7 ± 3.1%
	Post-package		Amber/CP laminated paper	100.0%	101.0 ± 1.4%	99.8 ± 3.1%	100.1 ± 4.7%	99.6 ± 5.4%
III	Classical	25 ± 2 °C/60 ± 5% RH	Amber/PC bottle	100.0%	100.4 ± 3.2%	99.1 ± 4.8%	99.0 ± 1.6%	99.6 ± 2.7%
	In-use		Amber/PC bottle	100.0%	98.9 ± 4.7%	98.9 ± 5.6%	99.1 ± 4.3%	98.7 ± 1.1%
	Post-package		Amber/CP laminated paper	100.0%	101.0 ± 3.4%	100.7 ± 1.3%	100.1 ± 4.7%	99.6 ± 2.4%
IV	Classical	25 ± 2 °C/60 ± 5% RH	Amber/PC bottle	100.0%	100.4 ± 3.2%	101.8 ± 4.8%	99.0 ± 1.6%	99.6 ± 4.7%
	In-use		Amber/PC bottle	100.0%	100.9 ± 4.9%	98.9 ± 4.7%	99.1 ± 3.5%	98.7 ± 3.3%
	Post-package		Amber/CP laminated paper	100.0%	101.0 ± 2.5%	100.7 ± 1.4%	100.1 ± 2.7%	99.6 ± 4.4%
V	Classical	25 ± 2 °C/60 ± 5% RH	Amber/PC bottle	100.0%	100.2 ± 1.2%	99.3 ± 2.8%	99.0 ± 1.5%	99.6 ± 2.7%
	In-use		Amber/PC bottle	100.0%	99.9 ± 2.5%	98.1 ± 4.8%	99.1 ± 2.5%	98.7 ± 4.1%
	Post-package		Amber/CP laminated paper	100.0%	101.0 ± 4.6%	99.4 ± 1.9%	100.1 ± 2.7%	99.6 ± 1.2%
Clonidine								
I	Classical	25 ± 2 °C/60 ± 5% RH	Amber/PC bottle	100.0%	100.2 ± 3.6%	99.8 ± 2.8%	99.0 ± 1.6%	99.6 ± 2.7%
	In-use		Amber/PC bottle	100.0%	99.8 ± 2.6%	98.9 ± 4.1%	99.1 ± 2.5%	98.7 ± 5.1%
	Post-package		Amber/CP laminated paper	100.0%	101.0 ± 2.2%	99.7 ± 1.1%	100.1 ± 2.7%	99.6 ± 1.2%
II	Classical	25 ± 2 °C/60 ± 5% RH	Amber/PC bottle	100.0%	100.2 ± 3.1%	99.8 ± 1.8%	99.0 ± 1.5%	99.6 ± 2.7%
	In-use		Amber/PC bottle	100.0%	99.1 ± 1.9%	98.9 ± 3.2%	99.1 ± 2.5%	98.1 ± 3.1%
	Post-package		Amber/CP laminated paper	100.0%	101.0 ± 2.2%	99.1 ± 2.4%	100.1 ± 2.1%	99.6 ± 1.2%
III	Classical	25 ± 2 °C/60 ± 5% RH	Amber/PC bottle	100.0%	100.8 ± 1.7%	99.8 ± 4.8%	99.0 ± 1.7%	99.6 ± 2.1%
	In-use		Amber/PC bottle	100.0%	99.7 ± 1.9%	98.9 ± 2.2%	99.1 ± 2.5%	98.1 ± 4.1%
	Post-package		Amber/CP laminated paper	100.0%	100.0 ± 3.2%	99.1 ± 3.3%	100.1 ± 2.1%	99.8 ± 1.2%
IV	Classical	25 ± 2 °C/60 ± 5% RH	Amber/PC bottle	100.0%	98.2 ± 1.6%	99.8 ± 1.8%	99.0 ± 4.1%	101.6 ± 2.1%
	In-use		Amber/PC bottle	100.0%	99.9 ± 1.1%	98.9 ± 4.0%	99.1 ± 2.5%	98.1 ± 5.0%
	Post-package		Amber/CP laminated paper	100.0%	101.0 ± 2.6%	99.1 ± 1.5%	100.5 ± 2.1%	102.6 ± 1.2%
V	Classical	25 ± 2 °C/60 ± 5% RH	Amber/PC bottle	100.0%	99.3 ± 3.7%	99.8 ± 2.8%	99.0 ± 1.6%	99.6 ± 2.1%
	In-use		Amber/PC bottle	100.0%	99.7 ± 3.9%	98.9 ± 4.1%	99.1 ± 2.4%	104.1 ± 5.1%
	Post-package		Amber/CP laminated paper	100.0%	101.0 ± 2.1%	99.1 ± 0.9%	100.5 ± 2.1%	99.6 ± 1.4%
Hydrocortisone								
I	Classical	25 ± 2 °C/60 ± 5% RH	Amber/PC bottle	100.0%	101.3 ± 3.2%	99.8 ± 2.8%	99.1 ± 2.7%	99.6 ± 2.1%
	In-use		Amber/PC bottle	100.0%	99.6 ± 2.9%	98.9 ± 2.1%	99.1 ± 1.5%	98.1 ± 1.1%
	Post-package		Amber/CP laminated paper	100.0%	101.5 ± 2.2%	99.1 ± 4.9%	100.5 ± 1.1%	101.6 ± 1.2%
II	Classical	25 ± 2 °C/60 ± 5% RH	Amber/PC bottle	100.0%	100.3 ± 3.3%	99.8 ± 1.8%	99.8 ± 1.6%	99.8 ± 1.5%
	In-use		Amber/PC bottle	100.0%	99.9 ± 1.9%	98.9 ± 2.7%	99.1 ± 2.5%	98.1 ± 6.1%
	Post-package		Amber/CP laminated paper	100.0%	100.5 ± 1.2%	99.1 ± 1.6%	101.5 ± 3.1%	98.6 ± 6.2%
III	Classical	25 ± 2 °C/60 ± 5% RH	Amber/PC bottle	100.0%	101.4 ± 2.7%	99.9 ± 2.4%	99.0 ± 1.9%	99.6 ± 2.1%
	In-use		Amber/PC bottle	100.0%	99.9 ± 1.9%	101.9 ± 4.3%	99.1 ± 2.7%	98.1 ± 2.1%
	Post-package		Amber/CP laminated paper	100.0%	101.0 ± 2.2%	99.1 ± 1.9%	100.1 ± 2.1%	99.6 ± 1.9%
IV	Classical	25 ± 2 °C/60 ± 5% RH	Amber/PC bottle	100.0%	101.3 ± 3.2%	99.8 ± 2.8%	99.7 ± 1.7%	99.6 ± 2.3%
	In-use		Amber/PC bottle	100.0%	99.4 ± 2.9%	98.9 ± 4.1%	99.4 ± 1.5%	98.1 ± 5.3%
	Post-package		Amber/CP laminated paper	100.0%	100.0 ± 1.2%	99.1 ± 1.3%	100.6 ± 2.9%	99.6 ± 2.2%
V	Classical	25 ± 2 °C/60 ± 5% RH	Amber/PC bottle	100.0%	100.3 ± 3.6%	99.8 ± 2.8%	99.0 ± 1.8%	99.6 ± 4.1%
	In-use		Amber/PC bottle	100.0%	99.9 ± 2.8%	98.9 ± 4.1%	99.1 ± 2.1%	98.1 ± 3.7%
	Post-package		Amber/CP laminated paper	100.0%	102.0 ± 1.2%	99.1 ± 1.5%	100.3 ± 2.7%	99.9 ± 1.7%

PC, polycarbonate; CP, cellophane and polyethylene. The initial clonidine concentrations (day 0) were presented as 100.0%.

**Table 4 children-10-01190-t004:** Results of uniformity test.

**Baclofen (10 mg/g, 5 mg/package)**														
**Facility**	**Sample**										**Uniformity**			
	**1**	**2**	**3**	**4**	**5**	**6**	**7**	**8**	**9**	**10**	**X**	**s**	**AV**	**Judgement**
I	107.3	103.9	95.7	96.6	95.5	106.9	98.5	102.2	92.6	94.3	99.3	5.3	12.8	Accept
II	101.0	107.4	100.8	102.3	94.2	96.6	99.4	92.9	105.2	104.2	100.4	4.7	11.3	Accept
III	95.5	101.2	94.8	95.0	106.6	104.7	95.8	97.7	105.7	100.0	99.7	4.6	11.1	Accept
IV	94.9	95.2	107.1	102.6	92.5	101.3	102.8	103.9	95.2	103.4	99.9	5.0	11.9	Accept
V	98.7	92.8	103.3	95.4	105.4	98.4	97.1	100.7	99.9	107.5	99.9	4.5	10.8	Accept
**Clonidine (0.2 mg/g, 0.1 mg/package)**														
**Facility**	**Sample**										**Uniformity**			
	**1**	**2**	**3**	**4**	**5**	**6**	**7**	**8**	**9**	**10**	**X**	**s**	**AV**	**Judgement**
I	101.8	96.1	105.4	98.7	106.5	107.0	98.1	102.5	106.5	104.9	102.7	4.0	9.5	Accept
II	94.7	105.0	106.4	99.7	103.1	100.9	97.6	96.6	97.1	96.1	99.7	4.0	9.6	Accept
III	96.6	105.9	98.5	96.2	105.8	98.9	96.0	93.0	98.6	102.4	99.2	4.3	10.2	Accept
IV	100.2	101.8	102.9	94.9	94.9	105.9	94.7	99.6	97.1	98.2	99.0	3.8	9.1	Accept
V	106.9	99.4	100.9	105.6	106.1	104.7	95.9	92.7	101.2	100.4	101.4	4.6	11.1	Accept
**Hydrocortisone (20 mg/g, 10 mg/package)**														
**Facility**	**Sample**										**Uniformity**			
	**1**	**2**	**3**	**4**	**5**	**6**	**7**	**8**	**9**	**10**	**X**	**s**	**AV**	**Judgement**
I	106.3	101.2	106.7	96.7	103.2	103.3	97.4	106.0	101.7	99.6	102.2	3.6	8.6	Accept
II	103.1	107.0	103.1	101.5	106.3	107.1	94.8	101.7	106.9	97.1	102.9	4.3	10.3	Accept
III	93.5	95.0	93.4	99.7	100.8	106.9	94.1	105.9	103.2	98.2	99.1	5.1	12.3	Accept
IV	99.8	105.6	105.2	103.1	102.9	102.2	106.3	95.0	102.7	96.9	102.0	3.7	8.9	Accept
V	100.6	95.8	105.0	102.1	93.6	99.6	95.8	100.8	102.8	102.1	99.8	3.6	8.7	Accept

X, mean of individual contents; s, sample standard deviation; AV, acceptance value. AV < 15.0 is acceptable value for the drug contents’ uniformity.

## Data Availability

Not applicable.

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
