# Peer review of "Study on the Preparation Method of Quality-Assured In-Hospital Drug Formulation for Children—A Multi-Institutional Collaborative Study"

_children, 2023, doi:10.3390/children10071190_

Round 1

Reviewer 1 Report

The authors have already published articles regarding the stability of baclofen, clonidine, and hydrocortisone in an oral powder form compounded for pediatric patients in Japan. The Reviewer would like to ask how the present manuscript differs from the others already published by the group.

Author Response

Thank you for your peer review.
We have attempted to make the corrections as attached.

Reviewer 2 Report

I have no comment on this manuscript. Manuscript is simple and straightforward. It has practical application.

Author Response

(The authors gave the same response as above.)

Reviewer 3 Report

This manuscript describes the preparation method of quality-assured in-hospital drug formulation. Quality assuring of in-hospital drug formulations is critical to achieving drug therapy from the viewpoints of efficacy and safety. The importance of obtaining data for quality assurance is understandable, but scientific papers must be based on a scientific rationale in addition to phenomenology.

1.       The manuscript is limited to describing experimental results, giving the impression that it is a different kind of manuscript from a scientific paper.

2.       The scientific novelty of this study is ambiguous. It is stated that the quality of in-hospital formulations in different environments and preparation methods is unknown, but what concerns are extracted? Physical stability, chemical stability, or variability in preparation among facilities? And will a method be proposed to scientifically resolve these concerns? Can the results of this study provide new criteria or standards for quality assurance?

3.       What does "Study methods" in Table 2 mean, and what does "Classical" mean?

4.       Data in Table 2 and Table 3 are not properly represented.

5.       The study only identified limited properties for three drugs. Therefore, it would be an overstatement to generalize as stated in the conclusions.

Author Response

(The authors gave the same response as above.)

Reviewer 4 Report

In the manuscript, the authors investigated how stability of drugs (baclofen, clonidine, hydrocortisone) can be changed when the commercially available dosage form is processed (decapsulation, tablet breaking, and milling, diluting) for pediatric therapy.
The structure of the manuscript is logical, although the introduction and conclusion could be more extended.

In the description of the preparation (2.1), the authors need to clarify some process parameters. For example, how long the milling process took? For sieving, it is necessary to specify particle size fraction. How long was the sieving time, for the sieve analyzer, what was the amplitude? Missing from the first table is how long was the mixing for the manual mixing process.

The manuscript can be be improved with giving additional information about the analytical method for determining the active substance content (e.g. eluent composition, type of columns, etc.)

Moderate checking and editing of English language is required

Author Response

(The authors gave the same response as above.)

Round 2

Reviewer 3 Report

The authors appropriately respond to reviewer's comments and suggestions.

I think this munuscript is acceptable for the publication.

Before the publication, two comments to be considered are as follows:

1. In table 3, the authors changed "Study method" to "Storage conditions", but "Storage methods" could be appropriate.

2. In table 4, the authors must write in English.
